# Microbial Prevalence and Antimicrobial Sensitivity in Equine Endometritis in Field Conditions

**DOI:** 10.3390/ani11051476

**Published:** 2021-05-20

**Authors:** María Luisa Díaz-Bertrana, Stefan Deleuze, Lidia Pitti Rios, Marc Yeste, Inmaculada Morales Fariña, Maria Montserrat Rivera del Alamo

**Affiliations:** 1Unit of Animal Medicine and Surgery, Veterinary Faculty, University of Las Palmas de Gran Canaria, ES-35416 Arucas, Spain; luigi.bertrana@ulpgc.es (M.L.D.-B.); inmaculada.morales@ulpgc.es (I.M.F.); 2Department of Clinical Science, Veterinary Faculty, University of Liege, B-4000 Liege, Belgium; s.deleuze@uliege.be; 3Equine Medicine Services, Veterinary Hospital, University of Las Palmas de Gran Canaria, ES-235416 Arucas, Spain; lpittirios@gmail.com; 4Biotechnology of Animal and Human Reproduction (TechnoSperm), Institute of Food and Agricultural Technology, University of Girona, ES-17003 Girona, Spain; marc.yeste@udg.edu; 5Department of Animal Medicine and Surgery, Veterinary Faculty, Universitat Autònoma de Barcelona, ES-08193 Bellaterra, Spain

**Keywords:** equine endometritis, early estrus, microbiologic study, sensitivity test

## Abstract

**Simple Summary:**

Endometritis diagnosis can be achieved by means of an endometrial biopsy and/or swab, the latter being used for cytology and microbiologic evaluation. Endometrial culturing plus a sensitivity test are crucial in infectious endometritis in order to determine the causal microorganism as well as the best antimicrobial treatment. In this study, endometrial swabs for culture and sensitivity test were obtained from 363 mares with reproductive failure. The most isolated microorganism was *Escherichia coli*, followed by *Staphylococcus* spp. and *Streptococcus* spp. Regarding sensitivity tests, the most efficient antibiotics were amikacin, cefoxitin and gentamicin, whereas cephaloridine and penicillin showed the lowest indexes. It can be concluded that, under the conditions of this study, β-lactam antibiotics are not efficient as a treatment for equine endometritis. In addition, microbiological and sensitivity studies are crucial to obtain good results when treating equine endometritis.

**Abstract:**

Endometritis is one of the main causes of infertility in mares. In the present study, 363 mares with a history of repetitive infertility, and positive endometrial cytology and/or vaginal discharge were included. An endometrial swab for microbiological purposes plus sensitivity test was obtained from each mare. A positive culture was obtained in 89% of mares. The main isolated genera were *Staphylococcus* (25.1%), *Streptococcus* (18.2%), *Escherichia* (17.3%) and *Pseudomonas* (12.1%). With regard to species, the most isolated microorganism was *Escherichia coli* (17.3%), *Staphylococcus* spp. (15.6%) and *Streptococcus* spp. (13.5%). Sensitivity tests showed that the most efficient antimicrobial was amikacin (57.3% of cultures), followed by cefoxitin (48.6%) and gentamicin (48.3%). When sensitivity test was analyzed in terms of Gram+ and Gram– bacteria, Gram+ were highly resistant to cephaloridine (77.3% of cultures), apramycin (70.8%) and penicillin (62.3%), whereas Gram– were highly resistant to penicillin (85.8%), followed by cephaloridine (78.9%). In conclusion, the present study shows the most prevalent microorganisms isolated from equine endometritis, which were found to be resistant to β-lactam antimicrobials. Likewise, these results highlight the significance of performing microbiological analyses as well as sensitivity tests prior to applying an antimicrobial therapy.

## 1. Introduction

Endometritis, both in its acute and chronic form, has long been recognised as one of the major causes of reduced fertility in the mare [1], thus being responsible for a severe economic impact on the equine breeding industry associated to failure to conceive and early embryonic death [2,3]. While endometritis has been associated with many causes, such as pneumovagina, urine pooling and the presence of semen in the uterine lumen, the most common aetiology is bacterial infection, mainly due to aerobic microorganisms [4].

The uterine lumen of mares has been usually assumed to be microorganism-free or to have transient non-resident microbiota [5]. However, recent studies suggest the presence of a normal microbiome, although its possible role in fertility has not been identified yet [6,7]. It is well established that high numbers of microorganisms can reach the uterus after parturition, since the cervix is wide open, and after mating [8]. After insemination, the uterus undergoes a local inflammatory response in the form of the infiltration of polymorphonuclear neutrophils (PMNs) [9], which remove microorganisms and excessive sperm cells from the uterine lumen [10]. In resistant mares, the inflammatory response of the endometrium is resolved within 48 h [11], whereas in those susceptible to endometritis, uterine clearance is delayed, allowing microorganisms to develop.

Bacterial species involved in infectious equine endometritis are usually residents of the normal microbiota of the mare, *Streptococcus equi* subspecies *zooepidemicus* being the most common bacteria inducing endometritis in the mare according to several studies [8,12,13,14,15]. However, other studies describe other microorganisms as common bacteria in equine endometritis, such as *Escherichia coli*, *Klebsiella pneumoniae*, *Pseudomonas* subspecies and *Staphylococcus aureus, Corynebacterium* spp., *Bacillus* spp, *Actinomyces* and *Lactobacillus* spp. [3,4,16,17]. Early diagnosis and proper instauration of antimicrobial treatment is crucial in the outcome of infectious endometritis. Thus, microbiological cultures and in vitro sensitivity tests are a prerequisite to establish the most adequate antibiotic therapy. However, delays in the initiation of the treatment and use of a broad-spectrum antibiotic, whose choice is based on the practitioner’s personal experience, are common in field conditions.

Instillation of the uterus with antibiotics is also a usual practice although it may, under some circumstances, be irritating on the endometrium and interfere with local defence mechanisms [2]. Previous studies demonstrated the efficiency of intrauterine antibiotic therapy to improve pregnancy rates when administered either before or after mating or AI [18,19].

Thus, the present study aimed at: a) determining infective microbiota in breeding mares with fertility problems by means of microbiological studies, and b) evaluating bacterial sensitivity to a set of antibiotics currently used to treat infertility in mares in field conditions.

## 2. Materials and Methods

### 2.1. Animals and Sampling

In the present study, a total of 363 mares from five different stud farms were included. Mares were 2 to 21 years old (mean age 11.5 years) and belonged to different breeds (Andalusian: 226; Crossbred: 109; English: 47; French Saddlebred: 2; Arabian: 1; Holstein: 1 and KWPN: 1). This is a retrospective study performed in mares for commercial purposes from AI centres. Inclusion criteria were history of repetitive infertility after two or more consecutive breedings/artificial inseminations with a stallion of proven fertility and a positive endometrial cytology [4] or presence of abnormal vaginal discharge. Mares were first sampled for endometrial culture and a second swab was obtained for endometrial cytology. Since all manipulations were clinically needed, permission from the Ethical Committee was not mandatory. A signed consent from owners to use data was obtained.

### 2.2. Endometrial Cytology

A uterine swab was obtained by means of a sterile double-guarded uterine culture swab (Equi-vet^®^, Madrid, Spain) during the oestrus phase. The swab was rolled on a slide which was allowed to air-dry. Then, the smear was fixed with methanol and further stained with May–Grünwald–Giemsa [20]. Ten fields at a magnification of 400× were evaluated for polymorphonuclear neutrophils (PMN) counting. A positive diagnosis for endometritis was considered when a mean value above two PMN/field was obtained [21].

### 2.3. Samples Processing

After careful scrub of external genitalia with 1% povidone iodine, a sterile double-guarded uterine culture swab (Equi-vet^®^, Madrid, Spain) was passed through the open cervix of the oestrous mares. Samples were all obtained by the same person. Swabs were transported to the laboratory at room temperature in a SP4 II culture medium without ampicillin [22] to be analysed. Samples with evident colour change of the SP4 II media, indicating bacterial growth, were inoculated in the different culture media described below. Those that showed no colour change were incubated at 37 °C for at least 24 h, and inoculated in the different media when colour change was evident. Inoculated plates were incubated for 24 h. Those plates with no growth of colonies were incubated for another 24 h. Identification of colonies and sensitivity tests were performed after incubation. Microbiological and sensitivity tests were performed at Epidemiology and Preventive Medicine Laboratory of the veterinary college of Las Palmas de Gran Canaria University

### 2.4. Culture Media

Samples were cultured in Columbia Agar, MacConkey Agar, Glucose Sabouraud Agar and Baird–Parker Agar to allow as many of the potentially present microorganisms to grow. All media were prepared at the Epidemiology and Preventive Medicine Laboratory, Veterinary College (Las Palmas de Gran Canaria University, Spain). Reagents for preparing culture media were all purchased from Panreac Química S.A. (Barcelona, Spain).

### 2.5. Identification of Colonies

Identification of colonies was initially performed by means of Gram staining and biochemical tests; specifically: catalase test, coagulase test and oxidase test. Then, API tests (bioMérieux S.A., France) were also performed. Two API tests were included in the present study. On the one hand, the API Staph was used to identify *Staphylococcus* spp., *Micrococcus* spp. and *Kocuria* spp. On the other hand, API 20E was used to identify microorganisms belonging to the genera *Enterobacteriaceae* and other non-fastidious Gram-negative rods.

### 2.6. Antimicrobial Susceptibility

Once colonies were isolated and identified, a disk diffusion antibiotic sensitivity test was performed in either Mueller Hinton broth or blood agar in the case of streptococci. Once microorganisms were inoculated, plates were incubated at 37 °C for 24 h. Bacterial growth inhibition was then evaluated and the results categorised as resistant, intermediate or sensitive according to the guidelines of the Clinical and Laboratory Standards Institute (CLSI). Diffusion disks were purchased from Oxoid S.A. (Madrid, Spain).

Tested antibiotics were selected based on previous combined data of efficacy against uterine infections and harmlessness on endometrium [23]. Thus, selected antibiotics were amikacin (B. Braun Medical S.A., Terrassa, Spain), ampicillin (Antibióticos de México, Mexico DF, Mexico), apramycin (Elanco Valquímica S.A, Alcobendas, Spain), gentamicin (B. Braun Medical S.A., Terrassa, Spain), kanamycin (Bristol-Myers Squibb, Madrid, Spain), penicillin (Fort Dodge Pfizer, Olot, Spain), neomycin (AB Biodisk, Solna, Sweden), ticarcillin (Oxoid, Limited, Hampshire, United Kingdom), cefoxitin (Oxoid, Limited, Hampshire, United Kingdom), cephaloridine (Oxoid, Limited, Hampshire, United Kingdom), oxytetracycline (AB Biodisk, Solna, Sweden), doxycycline (Oxoid, Limited, Hampshire, United Kingdom), amoxicillin/clavulanic acid (Oxoid, Limited, Hampshire, United Kingdom) and trimethoprim-sulphonamide (Grappiolo, Milan, Italy).

Sensitivity tests were performed on a total of 347 isolates.

### 2.7. Statistical Analyses

Data were analysed with a statistical package (IBM^®^ SPSS^®^ for Windows 25.0; Armonk, NY, USA). Response of each bacterium to each of the 14 antibiotics was graded as resistant, intermediate and sensitive. Proportions of each of these three categories were compared through a chi-square test and Z-test with Bonferroni correction. The level of significance was set at *p* ≤ 0.05.

## 3. Results

### 3.1. Identification of Colonies

Microorganisms were isolated from 323 out of the 363 mares (89.0%) included in the study. A total of 347 isolates were obtained from the 323 positive mares. Yeasts were identified in three isolates. Gram-positive and -negative bacteria were observed in 155 and 189 isolates, respectively (Figure 1).

The majority of uterine samples (93.2%, 301 of 323 mares) yielded a pure culture. Only 22 mares yielded mixed growth. Staphylococci were isolated in 20 mares out of these 22 mares. *Escherichia coli* was isolated in eight out of the 22 mares and was always combined with staphylococci (Table 1).

Regarding oxygen requirements, the most common microorganisms were facultative anaerobic bacteria (83.4%), followed by strict aerobic bacteria (16.0%). The only facultative aerobic species detected was *Agrobacterium radiobacter*, which was only obtained in two positive cultures (0.7%).

When bacterial genera identification was performed, the most isolated genera were *Staphylococcus*, in 25.1% of the isolates, followed by *Escherichia*, in 17.3% of isolates (Table 2). Other frequent genera were *Streptococcus* and *Pseudomonas,* which were obtained in 14.1% and 12.1% of isolates respectively (Table 2).

Finally, when species identification was performed, the most frequently isolated microorganism was *Escherichia coli,* being observed in 60 mares (17.3%), followed by *Staphylococcus* spp. and *Streptococcus* spp. non-haemolytic (15.6% and 13.5% respectively) (Table 2).

### 3.2. Antimicrobial Susceptibility

When results from antimicrobial susceptibility were evaluated, the most efficient antibiotics were amikacin, cefoxitin and gentamicin. Overall, all combined bacteria were sensitive in 57.3%, 48.6% and 48.3% of the evaluated samples, respectively (Table 3). On the other hand, microorganisms were significantly (*p* < 0.05) more resistant to cephaloridine and penicillin, specifically in 78.2% and 75.2% of the evaluated yielded bacteria (Table 3).

When antimicrobial susceptibility was analysed in terms of Gram+ and Gram– microorganisms, Gram+ bacteria were more sensitive to gentamicin and amikacin (44.8% and 46.8% respectively) and were resistant to ticarcillin (45.5%), trimethoprim-sulphonamide (47.1%), doxycycline (48.7%), kanamycin (51.3%), ampicillin (53.2%), penicillin (62.3%), apramycin (70.8%) and cephaloridine (77.3%) (Figure 2).

Regarding Gram– microorganisms, they showed the highest sensitivity to gentamicin (51.1%) and amikacin (65.8%). However, Gram– bacteria were resistant to most of the tested antibiotics. Thus, they showed to be resistant to doxycycline (57.4%), ticarcillin (59.5%), apramycin (60.0%), amoxicillin/clavulanic acid (66.3%), ampicillin (68.9%), cephaloridine (78.9%) and penicillin (85.8%) (Figure 3).

Individual sensitivities of each antibiotic evaluated according to the most frequently yielded bacteria are shown in Table 4. Briefly, *E. coli* was significantly (*p* < 0.05) sensitive to gentamicin, amikacin, ampicillin, kanamycin and cefoxitin, whereas it was significantly (*p* < 0.05) resistant to doxycycline and cephaloridine. *Staphylococcus* spp. was significantly sensitive to amikacin and amoxicillin/clavulanic acid and resistant to apramycin and cephaloridine. Regarding to *Streptococcus* spp., this microorganism was significantly sensitive to cefoxitin and resistant to apramycin and cephaloridine. Finally, *P. aeruginosa* was sensitive to kanamycin, trimethoprim-sulphonamide and cefoxitin, whereas it was significantly resistant to penicillin, doxycycline and cephaloridine.

## 4. Discussion

Bacterial endometritis has been reported to be present in 25% to 60% of barren mares [4,17,24,25,26,27]. In the present study, almost 90% of the mares yielded a positive culture. This important difference in the percentage of prevalence is probably due to inclusion criteria. Since in the present study all the included mares showed history of infertility and had a positive endometrial cytology, a higher percentage of positive mares was expectable.

On the other hand, the higher variability of culture media used in the study could also play some role. In this sense, Riddle et al. [4] and Davis et al. [17] used blood and Levine Eosin-Methylene Blue agar plates. Blood agar is a growth medium that facilitates the growth of fastidious microorganisms such as streptococci, whereas Levine growth medium is specific for *Escherichia coli*, *Enterobacter*, coagulase+ staphylococci and *Candida albicans*. In the case of our study, the media utilised were Columbia, McConkey, Glucose-Sabouraud and Baird Parker agars. Columbia agar facilitates the growth of streptococci species. McConkey agar allows the selection and recovery of Gram– bacilli and inhibits the growth of Gram+ microorganisms. Regarding glucose-Sabouraud agar, it provides support to the growth of fungal organisms and yeasts. Finally, Baird Parker agar facilitates the growth of staphylococci species.

Some mares showed positive cytology but negative microbial culture. The most plausible explanation for this observation would be non-infectious endometritis induced by an unresolved inflammation. Other reasons would include limitations of the sampling technique and presence of biofilms produced by some microorganisms. Focusing on the sampling technique, the sensitivity of different methods (namely double guarded cotton swab, uterine cytobrush and uterine biopsy) has been evaluated [28]. The less sensitive sampling technique is double guarded cotton swab, while endometrial cytobrush and biopsy have been reported to yield similar results [29,30,31]. The lower sensitivity of cotton swabs can be explained by the fact that routine swabbing only samples the most superficial layers of the endometrium and, on the other hand, because focal infections could be missed, as only a small area of the endometrium is sampled [32].

Another important issue are biofilms since, most often, diagnosing biofilm-producing bacteria by means of routine microbiological culture techniques is difficult [33], meaning that negative cultures may actually be positive. Several biofilm-producing microorganisms have been described to be involved in persistent infectious diseases (see [34] for a review). Focusing on equine endometritis, *Escherichia coli*, *Pseudomonas aeruginosa* and *Klebsiella pneumoniae* isolated from uterine samples were hypothesized to produce biofilms, thus being the most feasible cause of non-resolving and/or chronic endometritis after repeated treatments with antimicrobial therapy [2,35]. In fact, the ability of *P. aeruginosa* of producing biofilms in equine endometritis has been experimentally demonstrated in other studies [36].

Isolated bacteria in equine endometritis are highly variable depending on the study. In general terms, the most common isolated bacteria described in the literature are *Streptococcus equi* subspecies *zooepidemicus* and *Escherichia coli* [4,16,30,37]. Other bacteria described as causal agents of endometritis are *Pseudomonas aeruginosa*, *Klebsiella pneumoniae* and *Staphylococcus* spp [38]. In the present study, the most isolated microorganism was *Staphylococcus* spp. followed by *E. coli*, whereas *Streptococcus equi* subspecies *zooepidemicus* was isolated from two mares only. These differences in isolated microorganisms have been previously suggested to be due to the different geographic locations of the mares included in the studies, the different population of sampled mares and the exposure to different antimicrobial drugs [17]. On the other hand, the low prevalence of *Streptococcus equi* subspecies *zooepidemicus* could be explained by the sampling technique [39]. As aforementioned, routine swabbing for diagnosing endometritis only samples the most superficial layers of the endometrium. However, *Streptococcus equi* subspecies *zooepidemicus* provokes deeper infections [40] and, consequently, the present results may be underestimating the prevalence of this specific microorganism as it has been previously suggested [39]. However, it is worth noting that *Streptococcus equi* subspecies *zooepidemicus* has been reported as the main microorganism involved in endometritis already in the late 1970s [26], when endometrial sampling was performed by means of routine swabbing. This reinforces the hypothesis that microorganisms isolated from equine uterus widely vary between geographical areas.

Another factor to take into consideration is the existing differences among microbiology laboratories protocols. Many factors can affect the final result of a bacterial culture such as sampling method, transport conditions to the laboratory, sampling conservation until inoculated in Petri dishes, used culture media and time and conditions of incubation. Thus, comparison between studies may be sometimes hazardous. In this sense, it is worth mentioning that our laboratory was not always able to reach the category of species for every isolated bacterium, which would maybe partially modify the percentage of prevalence mainly in *Staphylococcus* and *Streptococcus* genera.

*Escherichia coli*, *Staphylococcus aureus* and α-haemolytic streptococci, which are normal constituents of the microbiota of the mare genital tract, residing mainly in the vaginal vestibule and the clitoral fossa, are considered as opportunistic microorganisms [5,41]. The presence of these microorganisms could also be considered as sample contamination. However, since all the mares included in the present study showed a positive endometrial cytology or abnormal vaginal discharge, it can be assumed that the positive growth was representative of the uterine infection. The present results confirm that the microorganisms involved in equine endometritis highly vary depending on the circumstances, highlighting the importance of performing an appropriate bacteriological culture before making any therapy decision in equine endometritis.

Another issue that needs to be discussed is the proportion of pure vs. mixt cultures. In the present study, more than 90% of the mares yielded a pure culture. These results are similar to those obtained by Davis et al. [17], who reported pure cultures in 87.4% of the analysed samples. When it comes to the 22 samples that yielded mixt cultures, the most frequent associated microorganisms belonged to the genera *Staphylococcus* (20/22), followed by *Escherichia coli* (8/22) that was always associated with staphylococci. These results are not fully concordant with those of Davis et al. [17], who observed that the microorganism yielded more frequently in combination was *Escherichia coli*. A possible explanation for this difference would be the higher prevalence of staphylococci in our study and the difference number of samples between our study and that of Davis et al. [17].

In field conditions, antibiotic therapy for endometritis is often selected empirically according to the personal experience of the veterinarian or based on previous studies, rather than after performing a sensitivity test [38]. However, this is not the most appropriate approach for endometritis treatment even in field conditions. According to sensitivity tests in the present study, the most efficacious antibiotic was amikacin, followed by gentamicin. Our results are partially in agreement with those of Albihn et al. [42], who observed that the most efficacious antibiotic, in general terms, was gentamicin. However, more recent reports from USA have observed general higher efficacies for trimethoprim-sulphonamide (>90%), followed by amikacin (>80%) [43].

On the other hand, the response to antibiotic therapy varies when different microorganisms are taken into consideration. Comparing sensitivity results among studies is a difficult exercise since results are affected by geographical locations or even antibiotic treatment policies. Thus, Gram+ microorganisms showed the highest sensitivity to amikacin followed by gentamicin, whereas Gram– microorganisms showed the highest sensitivity to amikacin, followed by cefoxitin and gentamicin. Going in depth, species-specific variations have been observed. Thus, while amikacin has been demonstrated to be an efficacious antibiotic in the most common isolated bacteria, gentamicin shows to be efficient for *E. coli*, *Streptococcus* spp. and *Pseudomonas aeruginosa*, but not in infections induced by *Staphylococcus* spp. Focusing on *P. aeruginosa* infections, this microorganism was highly sensitive to antibiotics such as trimethoprim-sulphonamide and cefoxitin, which did not show high efficiency in general terms. The present results agree with those recently reported by Pisello et al. [39], who showed that the most efficacious antimicrobial drugs are amikacin and gentamicin, together with marbofloxacin in *E. coli*-induced endometritis, whereas *Streptococcus equi* subspecies *zooepidemicus* showed higher sensitivities to ceftiofur, penicillin, rifampin and thiamphenicol. Thus, studies, both previous and current, outline the importance of performing microbiological and sensitivity tests to appropriately treat bacterial endometritis in the mare instead of advocating for blind treatments based on personal experience.

It is worth noting that sensitivity test results showed low percentages of efficacy in the present study. General results show that sensitivity percentages are below 60%. Thus, in addition to bacterial sensitivity to antimicrobial drugs, bacterial resistance to these drugs is also relevant and needs to be taken into consideration. In the present study, antibacterial efficacy of 14 antibiotics was tested. Isolated microorganisms in this study showed a resistance above 50% to half of the tested antibiotics. The main reason for the decrease in antibiotic efficacy has been associated to their inappropriate use, either by over-prescription, overuse or inadequate following of the antibiotic course. Obviously, blind treatments may also increase the resistance to antimicrobial drugs, being then inadvisable.

In general terms, the less efficient antimicrobial drugs were cephaloridine and penicillin. Regarding penicillin, bacterial sensitivity to this antimicrobial drug has progressively decreased over the years. Thus, while Albihn et al. [42] reported a percentage of sensitivity to penicillin of 86% in 2003, Pisello et al. [39] reduced that figure to 8.4% in 2019. In the present study, susceptibility to penicillin ranged from 0% to 38.3% depending on the isolated microorganism. As expected, Gram– bacteria were more resistant to penicillin than Gram+ (85.8% vs 62.3% respectively). However, this difference between Gram+ and Gram– bacteria was not observed in terms of resistance to cephaloridine, showing a percentage of resistance of 78.9% and 77.3%, respectively. Since cephaloridine is also a β-lactam antibiotic, such a high percentage of resistance was expectable up to some extent.

β-lactams antimicrobials have been demonstrated to be quite effective against *Streptococcus equi* subspecies *zooepidemicus* [43]. Considering that in the present study, this specific microorganism has been isolated in only 0.6% of the samples, this could be another explanation for the low sensitivity of β-lactams antibiotics observed in our results.

On the other hand, it is important to bear in mind that the present results are based exclusively on laboratory results and no in vivo study was performed. It is worth mentioning that the use of intrauterine antibiotics has been both advocated and criticised in the literature. Some authors indicated that intrauterine administration of antibiotics reaches higher inhibitory concentrations than those administered systemically [44], whereas others advocated for systemic administration [3,45]. Unfortunately, studies on the efficacy of systemic antimicrobials in equine endometritis are still scarce and further research is warranted.

An important limitation of the present study is the absence of data collected from mares after performing intrauterine antibiotic therapy. Since this study was performed on privately owned mares and in field conditions, details about treatment were not correctly recorded. For that reason, this research was only focused on microbiological and sensitivity tests. Thus, further research investigating in vivo efficiency outcomes is warranted.

## 5. Conclusions

In conclusion, the present study highlights the importance of performing a microbiological study combined with sensitivity tests to better determine, on the one hand, the actual responsible microorganism of endometritis in the mare and, on the other hand, the most appropriate therapy in equine infectious endometritis. The high variability in yielded microorganisms observed in the literature, as well as the high resistance to antibiotic drugs, reinforce the idea that blind treatments for equine endometritis should be avoided.

## Figures and Tables

**Figure 1 animals-11-01476-f001:**
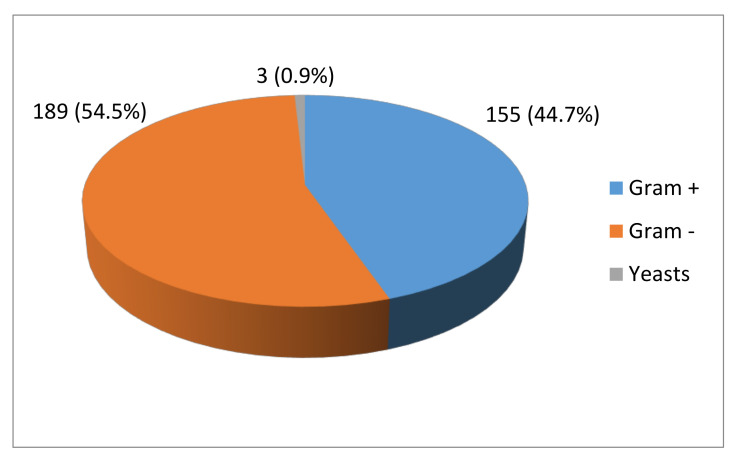
Percentage of Gram+, Gram– and yeasts in the 347 isolates obtained from the 323 positive mares included in the study.

**Figure 2 animals-11-01476-f002:**
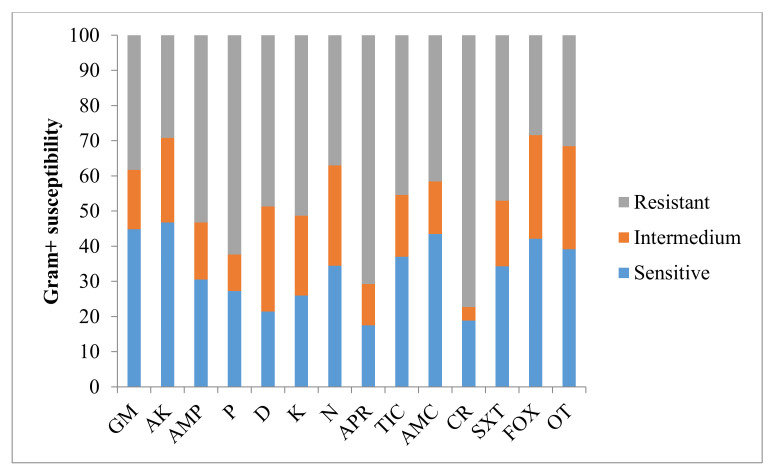
Antimicrobial susceptibility for Gram+ microorganisms. Results are expressed as percentages. GM: gentamicin; AK: amikacin; AMP: ampicillin; P: penicillin; D: doxycycline; K: kanamycin; N: neomycin; APR: apramycin; TIC: ticarcillin; AMC: amoxicillin/clavulanic acid; CR: cephaloridine; SXT: trimethoprim-sulphonamide; FOX: cefoxitin; OT: oxytetracycline.

**Figure 3 animals-11-01476-f003:**
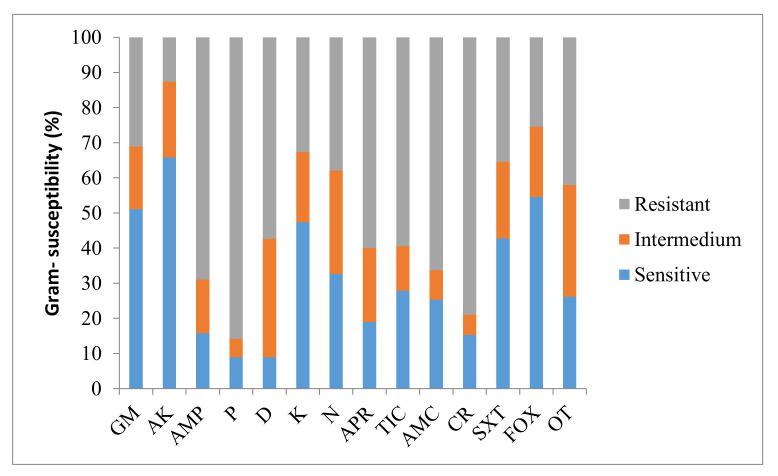
Antimicrobial susceptibility for Gram– microorganisms. Results are expressed in terms of percentage. GM: gentamicin; AK: amikacin; AMP: ampicillin; P: penicillin; D: doxycycline; K: kanamycin; N: neomycin; APR: apramycin; TIC: ticarcillin; AMC: amoxicillin/clavulanic acid; CR: cephaloridine; SXT: trimethoprim-sulphonamide; FOX: cefoxitin; OT: oxytetracycline.

**Table 1 animals-11-01476-t001:** Combinations of yielded mixed growth.

Microorganisms Combination	Number of Mares (out of 22)
*Staphylococcus + Escherichia coli*	8
*Staphylococcus + Pseudomonas*	4
*Staphylococcus + Klebsiella*	3
*Staphylococcus + Aeromonas*	1
*Staphylococcus + Proteus*	1
*Staphylococcus + Serratia*	1
*Staphylococcus + Streptococcus*	1
*Staphylococcus + Myroides*	1
*Enterococcus + Klebsiella*	1
*Micrococcus + Proteus*	1

**Table 2 animals-11-01476-t002:** Prevalence of bacteriological examinations yielded from uterine swabs.

Microorganism	Number of Isolates	Frequency (%)
*Escherichia coli*	60	17.3
*Staphylococcus* spp.	54	15.6
*Streptococcus* spp.	47	13.5
*Pseudomonas aeruginosa*	23	6.6
*Pseudomonas* spp.	19	5.5
*Klebsiella pneumoniae*	16	4.6
*Enterobacter aerogenes*	12	3.5
*Staphylococcus xylosus*	11	3.2
*Aerococcus viridans*	8	2.3
*Klebsiella ornithinolytica*	8	2.3
*Proteus* spp.	8	2.3
*Serratia* spp.	8	2.3
*Enterococcus faecalis*	7	2.0
*Enterobacter* spp.	6	1.7
*Staphylococcus epidermidis*	6	1.7
*Klebsiella* spp.	5	1.4
*Staphylococcus haemolyticus*	5	1.4
*Citrobacter* spp.	4	1.2
*Staphylococcus capitis*	4	1.2
*Staphylococcus lentus*	4	1.2
*Aeromonas hydrophila*	3	0.9
*Kluveria* spp.	3	0.9
*Micrococcus* spp.	3	0.9
*Proteus mirabilis*	3	0.9
*Agrobacterium radiobacter*	2	0.6
*Bordetella* spp.	2	0.6
*Candida* spp.	2	0.6
*Myroides* spp.	2	0.6
*Ochrobactrum anthropi*	2	0.6
*Staphylococcus intermedius*	2	0.6
*Streptococcus equi zooepidemicus*	2	0.6
*Candida tropicalis*	1	0.3
*Enterobacter sakazakii*	1	0.3
*Proteus panneri*	1	0.3
*Serratia odorifera*	1	0.3
*Staphylococcus lugdunensis*	1	0.3
*Vibrio parahaemolyticus*	1	0.3

**Table 3 animals-11-01476-t003:** Overall results of sensitivity to each evaluated antibiotic. Statistically significant differences (*p* < 0.05) among the response to sensitivity test for each antimicrobial drug are marked with different superscripts.

Antibiotic	Sensitive (%)	Intermediate (%)	Resistant (%)
Amikacin	57.3 ^a^	22.7 ^b^	20.0 ^c^
Cefoxitin	48.6 ^a^	24.5 ^a^	26.9 ^b^
Gentamicin	48.3 ^a^	17.4 ^b^	34.3 ^b^
Trimethoprim/sulphonamide	38.7 ^ab^	20.3 ^a^	41.0 ^b^
Kanamycin	37.8 ^ab^	21.2 ^a^	41.0 ^b^
Neomycin	33.4 ^a^	29.1 ^b^	37.5 ^c^
Amoxicillin/clavulanic acid	33.4 ^a^	11.3 ^b^	55.3 ^a^
Oxytetracycline	32.8 ^a^	30.6 ^b^	36.7 ^a^
Ticarcillin	32.0 ^a^	14.8 ^a^	53.2 ^a^
Ampicillin	22.4 ^a^	15.7 ^a^	61.9 ^b^
Apramycin	18.3 ^a^	16.9 ^b^	64.8 ^c^
Penicillin	17.2 ^a^	7.6 ^a^	75.2 ^b^
Cephaloridine	16.9 ^a^	4.9 ^b^	78.2 ^c^
Doxycycline	14.5 _a_	32.0 ^b^	53.5 ^c^

**Table 4 animals-11-01476-t004:** Sensitivity results of the different antibiotic drugs, expressed in percentages, for the main yielded microorganisms. Statistically significant differences (*p* < 0.05) among the response to sensitivity test for each microorganism and antimicrobial drug are marked with different superscripts.

	*E. coli*			*Staphylococcus* spp.			*Streptococcus* spp.			*P. aeruginosa*		
	S	IM	R	S	IM	R	S	IM	R	S	IM	R
GM	53.3 ^a^	20.0 ^ab^	26.7 ^b^	40.7 ^a^	18.5 ^a^	40.7 ^a^	44.7 ^a^	12.8 ^a^	42.6 ^a^	56.5 ^a^	21.7 ^a^	21.7 ^a^
AK	63.3 ^a^	28.3 ^a^	8.3 ^b^	48.1 ^a^	24.1 ^ab^	27.8 ^b^	46.8 ^a^	25.5 ^a^	27.7 ^a^	78.3 ^a^	17.4 ^ab^	4.3 ^ab^
AMP	16.7 ^a^	20.0 ^ab^	63.3 ^b^	38.9 ^a^	13.0 ^a^	48.1 ^a^	25.5 ^a^	19.1 ^a^	55.3 ^a^	17.4 ^a^	13.0 ^a^	69.6 ^a^
P	13.3 ^a^	8.3 ^a^	78.3 ^b^	27.8 ^ab^	7.4 ^b^	64.8 ^a^	38.3 ^ab^	6.4 ^b^	55.3 ^a^	13.0 ^a^	4.3 ^a^	82.6 ^b^
D	5.0 ^a^	33.3 ^b^	61.7 ^b^	18.5 ^a^	24.1 ^a^	57.4 ^a^	17.0 ^a^	36.2 ^b^	46.8 ^ab^	0.0 ^a^	39.1 ^b^	60.9 ^b^
K	41.7 ^a^	23.3 ^ab^	35.0 ^b^	27.8 ^a^	20.4 ^a^	51.9 ^a^	38.3 ^a^	23.4 ^a^	38.3 ^a^	56.5 ^a^	13.0 ^ab^	30.4 ^b^
N	33.3 ^a^	23.3 ^a^	43.3 ^a^	44.4 ^a^	24.1 ^a^	31.5 ^a^	31.9 ^ab^	36.2 ^b^	31.9 ^a^	21.7 ^a^	52.2 ^b^	26.1 ^a^
APR	18.3 ^a^	20.0 ^a^	61.7 ^a^	24.1 ^a^	5.6 ^a^	70.4 ^b^	17.0 ^a^	19.1 ^ab^	63.8 ^b^	17.4 ^a^	17.4 ^a^	65.2 ^a^
TIC	28.3 ^a^	11.7 ^a^	60.0 ^a^	37.0 ^a^	18.5 ^a^	44.4 ^a^	44.7 ^a^	21.3 ^a^	34.0 ^a^	34.8 ^a^	13.0 ^a^	52.2 ^a^
AMC	23.3 ^a^	10.0 ^a^	66.7 ^a^	50.0 ^a^	7.4 ^ab^	42.6 ^b^	42.6 ^a^	23.4 ^a^	34.0 ^a^	21.7 ^a^	13.0 ^a^	65.2 ^a^
CR	15.0 ^a^	8.3 ^a^	76.7 ^b^	22.2 ^a^	3.7 ^a^	74.1 ^b^	14.9 ^a^	8.5 ^a^	76.6 ^b^	13.0 ^a^	0.0 ^a^	87.0 ^b^
SXT	36.1 ^a^	25.0 ^a^	38.9 ^a^	38.2 ^a^	20.6 ^a^	41.2 ^a^	34.3 ^a^	11.4 ^a^	54.3 ^a^	71.4 ^a^	7.1 ^ab^	21.4 ^b^
FOX	55.6 ^a^	19.4 ^ab^	25.0 ^b^	32.4 ^a^	38.2 ^b^	29.4 ^a^	41.4 a	25.7 ^ab^	22.9 ^b^	64.3 ^a^	21.4 ^ab^	14.3 ^b^
OT	33.3 ^a^	22.2 ^a^	44.4 ^a^	36.4 ^ab^	39.4 ^b^	24.2 ^a^	50.0 ^a^	17.6 ^a^	32.4 ^a^	15.4 ^ab^	53.8 ^b^	30.8 ^a^

GM: gentamicin; AK: amikacin; AMP: ampicillin; P: penicillin; D: doxycycline; K: kanamycin; N: neomycin; APR: apramycin; TIC: ticarcillin; AMC: amoxicillin/clavulanic acid; CR: cephaloridine; SXT: trimethoprim-sulphonamide; FOX: cefoxitin; OT: oxytetracycline.

## Data Availability

Data are available under request to the authors.

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
