# Peer review of "Microbial Prevalence and Antimicrobial Sensitivity in Equine Endometritis in Field Conditions"

_animals, 2021, doi:10.3390/ani11051476_

Round 1

Reviewer 1 Report

The authors tried to evaluate the endometrial cytology, microbiology and antimicrobial sensitivity of isolated bacteria in infertile mares. The topic is of high scientific significance and might be interesting to broad audience. Methodology is scientifically sound and overall organization of the manuscript is fine but the English language in general and technical language in specific needed to be corrected. Discussion is too long to stay focused and should be comprehended to minimum. Conclusions should be revised to give a clear message of the whole manuscript.

Few more comments:

Line 46: early fetal loss should be replaced by early embryonic death

Sections 2.3 and 2.4: How long did you incubate the cultures to grow? 

Lines 158: This is not a clear sentence. You need to mention for which growth type e.g. bacteria, fungus or something else?  

Overall it is a nice job!

Author Response

Reviewer 1

The authors tried to evaluate the endometrial cytology, microbiology and antimicrobial sensitivity of isolated bacteria in infertile mares. The topic is of high scientific significance and might be interesting to broad audience. Methodology is scientifically sound and overall organization of the manuscript is fine but the English language in general and technical language in specific needed to be corrected. Discussion is too long to stay focused and should be comprehended to minimum. Conclusions should be revised to give a clear message of the whole manuscript.

Answer: Conclusions have been revised and re-written. Regarding to shorten the discussion, it hasn’t been possible since the other reviewers has asked for modifications.

Few more comments:

Line 46: early fetal loss should be replaced by early embryonic death

Answer: the sentence has been modified according to your suggestion

Sections 2.3 and 2.4: How long did you incubate the cultures to grow? 

Answer: this information has been added to the text

Lines 158: This is not a clear sentence. You need to mention for which growth type e.g. bacteria, fungus or something else?  

Answer: Unfortunately, the authors are not sure about what the reviewer is asking. Hopefully, the next modification will answer your question: When bacterial genera identification was performed, the most isolated genera were Staphylococcus, in 25.1% of the isolates, followed by Streptococcus, in 18.2% of isolates (Table 3). Other frequent genera were Escherichia and Pseudomonas, which were obtained in 17.3% and 12.1% of isolates respectively (Table 3).

Overall it is a nice job!

Reviewer 2 Report

General comments

The authors conducted a large field study on endometrial cultures and antibiotic sensitivity patterns in broodmares from commercial local stud farms. The manuscript is well written and should be of interest to the reader. A major concern is however, associated with the presentation and discussion of the data. While the authors justifiably emphasize the importance of bacterial culture and sensitivity to specific bacteria to guide treatment of endometritis, data is also presented as overall antibiotic sensitivity and sensitivity to gram positive versus gram negative bacteria, with recommendations for “blind” treatment of endometritis without culture and sensitivity. Considering antibiotic resistance and an overuse (or non-specific use) of antibiotics as an emerging public health issue, this clinical practice should not be encouraged or suggested in a scientific publication. The authors are therefore, encouraged to reorganize data presentation and emphasize the antibiotic sensitivity results for specific bacterial species.    

Since prevalence of bacteria and antibiotic resistance are likely to be geographically linked based on environment and prevalence of antibiotic use, these circumstances should also be emphasized when the results are discussed.    

Specific comments:

Lines 45-48: Endometritis is one form of uterine infection, commonly observed in cycling mares and confined to the endometrium. The condition is as far as we know, not the cause of other forms of uterine infections associated with mid-term abortions, placentitis (infection of the placenta and fetus), neonatal sepsis, or post-partum metritis (infection in all layers of the uterus). Please omit or revise this statement.

Lines 51-52: There are some recent data suggesting the presence of a normal uterine microbiome in mares.

Lines 71-72L I suggest the sentence should read: “….usual practice although it may, under some circumstances be irritating on the endometrium and interfere with local defense mechanisms (2).”

Line 83: How many stud farms were included in the study?

Lines 92-108: Please clarify the order of sampling if endometrial cytology and culture samples were obtained on the same occasion, since one may affect the other.  

Line 169: Table 2 or 3?

Lines 169-170: Streptococcus was found in 18% of the positive cultures and E. Coli in 17.3% according to Table 3. Please clarify.

Lines 181-190: This may not be relevant if antibiotic treatment is guided by culture and sensitivity results.

Table 5: This table is misleading since it does not account for sensitivity patters for specific bacteria. Table 6 contains more important information to the reader and the manuscript should be focused on these results.

Line 229: Suggest “…history of infertility and had positive endometrial cytology, a higher percentage….”

Line 294: “…representative of uterine infection.”

Lines 310-312: This is not good practice and should not be encouraged in a scientific publication. The information is also limited to specific geographic locations.

Line 318: Focus on the aspect of your findings.

Lines 334-335. Resistance to more than 50% of the tested antibiotics is concerning and needs further elaboration in the discussion.

Lines 338-339: Is the discrepancy of these reports the result of change over time or different geographic locations with potential differences in antibiotic treatment policies?

Lines 346-347 and lines 364-366: See comment for lines 310-312.    

Line 471: Please correct the typo “LeBlamares”.

Author Response

Reviewer 2

The authors conducted a large field study on endometrial cultures and antibiotic sensitivity patterns in broodmares from commercial local stud farms. The manuscript is well written and should be of interest to the reader. A major concern is however, associated with the presentation and discussion of the data. While the authors justifiably emphasize the importance of bacterial culture and sensitivity to specific bacteria to guide treatment of endometritis, data is also presented as overall antibiotic sensitivity and sensitivity to gram positive versus gram negative bacteria, with recommendations for “blind” treatment of endometritis without culture and sensitivity. Considering antibiotic resistance and an overuse (or non-specific use) of antibiotics as an emerging public health issue, this clinical practice should not be encouraged or suggested in a scientific publication. The authors are therefore, encouraged to reorganize data presentation and emphasize the antibiotic sensitivity results for specific bacterial species.

Answer: thanks for your comments. We have followed your advice and have modified the manuscript accordingly. The chapter on “blind” treatment has been deleted from the manuscript since you are totally right when suggesting this approach should not be encouraged in scientific publications.    

Since prevalence of bacteria and antibiotic resistance are likely to be geographically linked based on environment and prevalence of antibiotic use, these circumstances should also be emphasized when the results are discussed.    

Answer: Thanks for your comment. According to literature, bacterial prevalence might be related to geographical distribution and it has been discussed in the text. However, antibiotic resistance is probably linked more to the use of specific antimicrobials than to geographical issues (see answer for lines 338-338 below), which has also been discussed in the text

Specific comments:

Lines 45-48: Endometritis is one form of uterine infection, commonly observed in cycling mares and confined to the endometrium. The condition is as far as we know, not the cause of other forms of uterine infections associated with mid-term abortions, placentitis (infection of the placenta and fetus), neonatal sepsis, or post-partum metritis (infection in all layers of the uterus). Please omit or revise this statement.

Answer: the sentence has been omitted following your advice

Lines 51-52: There are some recent data suggesting the presence of a normal uterine microbiome in mares.

Answer: after performing a search in pubmed, the authors have found no related manuscript. We would much appreciate if the reviewer could suggest a specific reference on this topic

Lines 71-72L I suggest the sentence should read: “….usual practice although it may, under some circumstances be irritating on the endometrium and interfere with local defense mechanisms (2).”

Answer: thanks for your comment. The sentence has been modified according to your suggestion

Line 83: How many stud farms were included in the study?

Answer: samples came from 5 different studs. This information has been added to the text.

Lines 92-108: Please clarify the order of sampling if endometrial cytology and culture samples were obtained on the same occasion, since one may affect the other.  

Answer: A first swab was obtained for microbiological purposes and a second one was obtained for cytology as it is recommended in the literature. It is also the routine approach for field conditions.

Line 169: Table 2 or 3?

Answer: you are totally right. It is Table 3. It has been amended in the text

Lines 169-170: Streptococcus was found in 18% of the positive cultures and E. Coli in 17.3% according to Table 3. Please clarify.

Answer: It was a mistake. The correct data are those of the table and the text has been modified accordingly

Lines 181-190: This may not be relevant if antibiotic treatment is guided by culture and sensitivity results.

Answer: definitely, as the reviewer states, antibiotic therapy has to be decided according to culture and sensitivity results. However, the authors consider that an overall view of resistance/susceptibility of antimicrobials regardless the microorganisms is interesting. In addition, it’s a way to encourage clinicians not to use antimicrobials blindly. That is why we have decided to keep it.

Table 5: This table is misleading since it does not account for sensitivity patters for specific bacteria. Table 6 contains more important information to the reader and the manuscript should be focused on these results.

Answer: Same comment than previous question. Table 5 gives a general vision of sensitivity, while table 6 exposes a more focused vision and we think these two tables are complementary.

Line 229: Suggest “…history of infertility and had positive endometrial cytology, a higher percentage….”

Answer: thanks for your suggestion. The sentence has been modified accordingly

Line 294: “…representative of uterine infection.”

Answer: the sentence has been modified according to your suggestion

Lines 310-312: This is not good practice and should not be encouraged in a scientific publication. The information is also limited to specific geographic locations.

Answer: your comment is totally correct. This sentence has been modified to avoid the “encouragement” of blind treatments

Line 318: Focus on the aspect of your findings.

Answer: Thanks for your suggestion. The authors have tired to focus on their own findings. However, comparing with previous studies has been considered necessary to contextualize the current results.

Lines 334-335. Resistance to more than 50% of the tested antibiotics is concerning and needs further elaboration in the discussion.

Answer: Thanks for your comment. Some discussion on this concern has been added to the text.

Lines 338-339: Is the discrepancy of these reports the result of change over time or different geographic locations with potential differences in antibiotic treatment policies?

Answer: the discrepancies can be due to several reason. Certainly, antibiotic treatment policies could be an explanation, since not every single drug is allowed to be used in every single country. On the other hand, the overuse or the inappropriate use of antibiotics may have also a role in these discrepancies. Another factor to be taken into consideration are also personal preferences. In front of a bacterial infection, some clinicians advocate for some specific antimicrobial, making resistance easier to appear as time goes by. A sentence (lines 336-337) has been added to the text to cover this query.

Lines 346-347 and lines 364-366: See comment for lines 310-312.

Answer: Regarding to lines 346-347, any comment about “blind treatment” for endometritis in mares has been removed from the text. Regarding to lines 364-366, the authors considered a limitation the fact that this study was performed with owned mares and complete recordings of the efficacy of the antimicrobial treatment are lacking. Since establishing the in vivo response is interesting, we considered important to highlight that fact. However, if the reviewer considers deleting these two sentences as a necessary requirement, they can be removed from the text.

Line 471: Please correct the typo “LeBlamares”.

Answer: the typo has been corrected

Round 2

Reviewer 2 Report

The authors have addressed a majority of my concerns, while some issues remain to be addressed.

Since prevalence of bacteria and antibiotic resistance are likely to be geographically linked based on environment and prevalence of antibiotic use, these circumstances should also be emphasized when the results are discussed.    

Answer: Thanks for your comment. According to literature, bacterial prevalence might be related to geographical distribution and it has been discussed in the text. However, antibiotic resistance is probably linked more to the use of specific antimicrobials than to geographical issues (see answer for lines 338-338 below), which has also been discussed in the text.

Reviewer’s response: My concerns are related to the local limitation of your observations, and my suggestion was to strengthen the importance and interest in these findings by discussing your results in light of previous reports from other parts of the world, such as the abstract by Mitchell et al from 2018 (J. Eq. Vet. Sci. 66 (114). This is particularly important when general recommendations are provided on selection of antibiotics in contrast to results of previous work. In addition, only 0.6 of the samples were positive for Strep equi Zooepidemicus (one of the major causes of equine endometritis in other reports), and the majority of Strep spp were non-hemolytic in your report. This may have affected the antibiotic sensitivity results, specifically the surprisingly high resistance against b-lactam antibiotics, since they still are quite effective against Strep equi zooepidemicus in many parts of the world.

A discussion in line with these points would strengthen the relevance of the manuscript and make justice to your research.    

Lines 23-24: A large volume of data from other parts of the world suggests that b-lactam antibiotics are effective against major causative bacteria involved in equine endometritis, so this general statement is misleading. Please add “under the condition of this study”.   

Lines 51-52: There are some recent data suggesting the presence of a normal uterine microbiome in mares.

Answer: after performing a search in pubmed, the authors have found no related manuscript. We would much appreciate if the reviewer could suggest a specific reference on this topic

Reviewer’s response:

Heil BA, Thompson SK, Kearns TA, Davolli GM, King G, Sones JL. Metagenetic characterization of the resident equine uterine microbiome using multiple techniques. J Equine Vet Sci 66: 111, 2018. doi:10.1016/j.jevs.2018.05.156

Holyoak GR, Lyman CC, Wieneke X, DeSilva U. The equine endometrial microbiome. Clin Therio 10: 273–277, 2018.

Heil BA, Paccamonti DL and Sones JL have also published a good review article on this topic: https://doi.org/10.1152/physiolgenomics.00045.2019

Lines 92-108: Please clarify the order of sampling if endometrial cytology and culture samples were obtained on the same occasion, since one may affect the other.  

Answer: A first swab was obtained for microbiological purposes and a second one was obtained for cytology as it is recommended in the literature. It is also the routine approach for field conditions.

Reviewer’s response: Please clarify the order of sampling in the text as requested.

Lines 171-177: It is somewhat confusing to the reader when you refer to a table on the prevalence of bacterial species when reporting bacterial genera. I suggest that you either report genera and species in separate tables or modify Table 2 to facilitate for the reader. In addition, Staphylococci appeared in 25.2% (not 25.1%), and Streptococci in 14.1% (rather than 18.2%; Strep spp, 13.5%; Strep equi zooepidemicus, 0.6%)) when the species are computed from the table.  Please clarify and make appropriate changes to the manuscript.

Line 182: I suggest “All combined bacteria were sensitive….”

Line 189: I believe Trimethoprim-sulphonmide had the most favorable sensitivity to endometritis (>90%) in the report by Mitchell et al, 2018. Please discuss the difference when you make recommendations on treatment.

Lines 316-318: Please discuss your results in comparison to those from Mitchell et al, 2018. I suggest you change the sentence on line 317 to read “…the most efficacious antibiotic was amikacin..”

Author Response

Appreciate reviewer, thanks for accepting to review again our manuscript. Here are our answers to your questions

Reviewer’s response: My concerns are related to the local limitation of your observations, and my suggestion was to strengthen the importance and interest in these findings by discussing your results in light of previous reports from other parts of the world, such as the abstract by Mitchell et al from 2018 (J. Eq. Vet. Sci. 66 (114). This is particularly important when general recommendations are provided on selection of antibiotics in contrast to results of previous work. In addition, only 0.6 of the samples were positive for Strep equi Zooepidemicus (one of the major causes of equine endometritis in other reports), and the majority of Strep spp were non-hemolytic in your report. This may have affected the antibiotic sensitivity results, specifically the surprisingly high resistance against b-lactam antibiotics, since they still are quite effective against Strep equi zooepidemicus in many parts of the world.

A discussion in line with these points would strengthen the relevance of the manuscript and make justice to your research.

Answer: thanks for your suggestion. You are right when stating that b-lactam antimicrobials are quite effective in other parts of the world. Unfortunately, in Spain, veterinarians have to face many times owners’ self-decisions that decide to treat their animals (not only mares) by themselves without the advice of a clinician. This is having awful consequences on antimicrobials efficacy. On the other hand, discussion has been modified following your suggestion on the variability of antimicrobials sensitivity results and the low prevalence for S equi zooepidemicus (lines 359-363).    

Lines 23-24: A large volume of data from other parts of the world suggests that b-lactam antibiotics are effective against major causative bacteria involved in equine endometritis, so this general statement is misleading. Please add “under the condition of this study”.   

Answer: thanks for your suggestion. The sentence has been modified accordingly.

Lines 51-52: There are some recent data suggesting the presence of a normal uterine microbiome in mares.

Answer: after performing a search in pubmed, the authors have found no related manuscript. We would much appreciate if the reviewer could suggest a specific reference on this topic

Reviewer’s response:

Heil BA, Thompson SK, Kearns TA, Davolli GM, King G, Sones JL. Metagenetic characterization of the resident equine uterine microbiome using multiple techniques. J Equine Vet Sci 66: 111, 2018. doi:10.1016/j.jevs.2018.05.156

Holyoak GR, Lyman CC, Wieneke X, DeSilva U. The equine endometrial microbiome. Clin Therio 10: 273–277, 2018.

Heil BA, Paccamonti DL and Sones JL have also published a good review article on this topic: https://doi.org/10.1152/physiolgenomics.00045.2019

Answer: thanks for providing the literature. Introduction has been modified accordingly.

Lines 92-108: Please clarify the order of sampling if endometrial cytology and culture samples were obtained on the same occasion, since one may affect the other.  

Answer: A first swab was obtained for microbiological purposes and a second one was obtained for cytology as it is recommended in the literature. It is also the routine approach for field conditions.

Reviewer’s response: Please clarify the order of sampling in the text as requested.

Answer: This information has been added following your request.

Lines 171-177: It is somewhat confusing to the reader when you refer to a table on the prevalence of bacterial species when reporting bacterial genera. I suggest that you either report genera and species in separate tables or modify Table 2 to facilitate for the reader. In addition, Staphylococci appeared in 25.2% (not 25.1%), and Streptococci in 14.1% (rather than 18.2%; Strep spp, 13.5%; Strep equi zooepidemicus, 0.6%)) when the species are computed from the table.  Please clarify and make appropriate changes to the manuscript.

Answer: The authors agree with your comment. In fact, in the previous version of the manuscript, table 2 included only genera, whereas table 3 (now table 2) included species. However, the academic editor showed some concern because in table 3 some of the microorganisms were expressed as spp since the laboratory was not able to reach the species category. For that reason, table 2 was suppressed under the other reviewers’ request. We would be happy to put it back if you consider it necessary to clarify the content of the manuscript. Regarding to the percentages, the reviewer is completely right and there was a mistake in the percentage of streptococci bacteria. This mistake has been amended in the text. Regarding to the percentage of staphylococci, the difference depends on how the number is reached. If percentages are directly combined, the final percentage is 25.2%. However, if the number of isolates are considered, then the percentage is 25.1%.

Line 182: I suggest “All combined bacteria were sensitive….”

Answer: the sentence has been modified according to your suggestion

Line 189: I believe Trimethoprim-sulphonmide had the most favorable sensitivity to endometritis (>90%) in the report by Mitchell et al, 2018. Please discuss the difference when you make recommendations on treatment.

Answer: this has been added to the discussion

Lines 316-318: Please discuss your results in comparison to those from Mitchell et al, 2018. I suggest you change the sentence on line 317 to read “…the most efficacious antibiotic was amikacin..”

Answer: the verb has been changed to was instead of is. Reference to the paper by Mitchell at al., (2018) on TMS has been also added.